# Trends in Mycotoxins Co-Occurrence in the Complete Feed for Farm Animals in Southern Romania During 2021–2024 Period

**DOI:** 10.3390/toxins17040201

**Published:** 2025-04-15

**Authors:** Valeria Cristina Bulgaru, Mihail Alexandru Gras, Aglaia Popa, Gina Cecilia Pistol, Ionelia Taranu, Daniela Eliza Marin

**Affiliations:** 1National Research-Development Institute for Animal Biology and Nutrition (IBNA), Calea Bucuresti 1, 077015 Balotesti, Romania; gras_mihai@yahoo.com (M.A.G.); gina.pistol@ibna.ro (G.C.P.); ionelia.taranu@ibna.ro (I.T.); daniela.marin@ibna.ro (D.E.M.); 2Faculty of Biotechnologies, University of Agronomic Sciences and Veterinary Medicine of Bucharest, 59 Marasti Blvd., 011464 Bucharest, Romania; aglaia_popa@yahoo.com

**Keywords:** mycotoxins, feed, poultry, pig, Southern Romania

## Abstract

Mycotoxins are common natural contaminants of crops and fruits, associated with negative effects on human and animal health. Currently, more than 300 mycotoxins have been identified, but data on their effects and their limits in feed and food are still inconsistent. The European Commission, by directive EC 574/2011, established regulations concerning the maximum limit allowed in farm animals’ feed for aflatoxins, but for all other mycotoxins there are only recommendations (EC 1319/2016) and there are no established limits. Considering their variety and toxic effects, but also the fact that not many details are yet known about the cumulative effects of co-contamination with various mycotoxins, it is necessary to monitor the evolution of their presence in animal feed. The aim of our study was to analyze for a four-year period (2021–2024) the concentrations of six mycotoxins (total aflatoxins-AFT, fumonisins-FB, deoxynivalenol-DON, zearalenone-ZEA, T2/HT2 and ochratoxin (A + B)-OTA), the most frequently encountered in the south area of Romania in poultry, piglets and pig’s complete feed. Our results showed that the maximum highest concentrations were 5.8 ppb for AFT, 4.7 ppm for FB, 1.9 ppm for DON, 62.8 ppb for ZEA, 32.1 ppb for T2/HT2 and 19.7 ppb for OTA irrespective of the type of feed. It should be noted that AFT and ZEA were identified in all samples during the entire monitored period, and the only mycotoxin that exceeded the guidance value was DON, for which the recommendation of 0.9 ppm for pig feed was exceeded. Recent studies demonstrated that sub-chronic and chronic exposure to low concentrations of mycotoxins and specially co-contamination is more common than acute exposure, being able to affect animal health over time by lowering the defense capacity, inducing inflammatory reactions and affecting intestinal health, which in the long term could have important economic consequences. Our survey study can provide important data showing the degree of contamination with mycotoxins in pig and poultry feed including the simultaneous presence of different mycotoxins in this complete feed.

## 1. Introduction

Filamentous fungi are able to create toxic secondary metabolites known as mycotoxins [1]. Over time, the existence of more than 300 mycotoxins has been discovered, some of them being constantly present in food and feed [2]. Currently, many of these mycotoxins are not fully characterized, and their toxicity and mechanisms of action are still being investigated, existing also concerns regarding the simultaneous effect of multiple mycotoxins and long-term exposure, even at low concentrations [3]. The CE regulations and recommendations setting maximum allowed levels for mycotoxins in feed have been developed based only on toxicological evidence from research that examined only individual mycotoxin exposures and effects. They do not take into account the cumulative effects of multiple mycotoxin exposures or chronic exposure to low concentrations. However, Grenier and Oswald (2011) [4] showed that long-term exposures to two or more mycotoxins could be antagonistic, additive, or synergistic with harmful effects. At the European Union level, except for aflatoxins (EC 574/2011), there are only recommendations (EC 1319/2016) regarding the maximum levels of mycotoxins allowed in feed for farm animals. Therefore, studies to establish the limits of mycotoxins in feed and food are needed to diminish the animal’s health issues and to reduce the economic losses caused annually by exposure to mycotoxins [5].

Aflatoxins (AFT), zearalenone (ZEA), deoxynivalenol (DON), ochratoxins (OTA), and fumonisins (FB) are the most common mycotoxins discovered in cereal crops like corn, wheat, and barley [6]. From a worldwide standpoint, aflatoxins (B1, B2, G1, G2) are thought to be the class of mycotoxins that cause the most concern due to their high toxicity. The long-term exposure of farm animals to sub-acute levels of AFBs has been associated with teratogenicity, lower feed efficiency, increased susceptibility to illness, liver lesions, and/or tumors, and decreased eggshell and carcass quality [7,8,9]. Also, FB, DON, and ZEA are thought to be the most significant *Fusarium* mycotoxin [10] taking into account the effects on animal health and the resulting financial loss. For example, DON or vomitoxin [11], is associated with diarrhea, emesis, nausea, anorexia, and gastrointestinal bleeding [12]. Studies show that prolonged exposure to low concentrations of DON can induce toxic effects, especially in pigs [13]. Severe illnesses including pulmonary edema in pigs [14] and leukoencephalomalacia in horses [15] are associated with feed contaminated with FB. Additionally, studies have shown that FB could have nephrotoxic, hepatotoxic, and immuno-suppressive effects [16]. Unlike pigs, data shows that poultry does not have an increased sensitivity to DON, but at high concentrations, it can affect gut integrity [17] leading to weight loss, reduced appetite, and increased susceptibility to bacterial infections [18,19].

In pigs, fertility issues and hyper estrogenic symptoms, such as vulva swelling and uterine enlargement, are the primary causes of ZEA exposure, but studies have shown that ZEA can also trigger the immune response and affect the functions of other systems like digestive, hepatic, or nephrological [20]. Other research on pigs has demonstrated that exposure to feed co-contaminated with low concentrations of DON (2 ppm), NIV (1,3 ppm), and ZEA (1.5 ppm) for a month can cause histological alterations in the colon, liver, and lymphoid organs, reducing also the piglets’ rate of weight increase [21]. Furthermore, in poultry, studies are showing that concomitant exposure to AFT (13 ppb) and OTA (1.8 ppb) caused necrotic lesions of the kidneys and liver but also induced anorexia and inadequate development [22]. Other studies in poultry have indicated the ability of DON and FB to induce immunological changes and increase the severity of chickens with coccidiosis [23]. Nephropathy outbreaks in pigs and poultry have been associated with OTA contamination, the kidney is known as the main target of this toxin. Moreover, OTA is linked to immunosuppression, slowed growth, and higher mortality rates. Even at safe dosages, in vitro studies have demonstrated that OTA can exacerbate intestinal barrier dysfunctions caused by DON [24]. Also, it has been documented that oral lesions in poultry and a weakened acquired immunological response in pigs can result from dietary exposure to T2 and HT2 [25].

Regarding the co-occurrence, AFB + FB, DON + ZEA, AFB + OTA, and FB + ZEA are the most frequent among the 127 mycotoxin combinations found in cereals and samples of derived cereal products [26]. Nevertheless, only a small amount of research reported the most frequent existing combinations, as well as the concentrations of the co-occurring mycotoxins or the percentage of co-contaminated samples. Given all the literature data, it is important to monitor how mycotoxins are becoming more and more prevalent in animal feed due to their wide range of toxic effects as well as the lack of knowledge on the combined effects of co-contamination with different mycotoxins.

Therefore, the aim of our research was to provide data on the occurrence and concentration levels of six major mycotoxins (AFT, FB, DON, ZEA, T2/HT, and OTA) that are mostly found in poultry, piglets, and pigs complete feed produced in a local factory (pilot station of the National Institute of Biology and Animal Nutrition, Balotesti) from the southern region of Romania and their co-occurrence over a four-year period (2021–2024). The feed analyzed was selected based on the demand in the feed market and the economic importance of the farm animal species for which they were manufactured. According to Eurostat (European Statistical System), Romania has a pig population of 3.27 million, of which 637,000 are piglets [27] and approximately 7.5 million chicks of mixed meat-laying breeds [28]. The main cereals (purchased from local feed producers) used as the raw materials of all complete feed formulations were corn (>65%) and soybean meal (>19%), the degree of variation in the contamination level of the analyzed feeds depended on the percentage of cereals in the feed formulation calculated to ensure the nutrient requirements for the species and age category of the farm animals.

This type of study is necessary to have an image of the mycotoxin contamination of complete feed available in Southern Romania. Moreover, recent research has shown that low-concentration sub-chronic and chronic exposure to mycotoxins, particularly co-contamination, is more common than acute exposure [29]. These exposures can have long-term effects on animal health by reducing defense capacity, triggering inflammatory reactions, and affecting intestinal health. These effects may also have significant economic implications; therefore, the feed quality is even more important.

## 2. Results

### 2.1. The Prevalence of Mycotoxins in the Complete Feed for Farm Animals During 2021–2024 Period

The results in Table 1 show that AFT and ZEA are almost ubiquitous throughout the four-year period, regardless of the type of complete feed studied. The minimum number of mycotoxins detected simultaneously in pigs’, sows’, and poultry complete feed was four. In the feed of piglets and gilts at least five mycotoxins were positive at the same time, AFT, DON, and ZEA being present in each tested sample, although the concentrations exceed the EC recommendation only in a small percentage for DON, the presence of a large number of mycotoxins could induce negative effects on the animal health, especially due to the sensitivity of piglets after weaning, which do not have a fully developed immune and digestive system.

### 2.2. The Evolution of Mycotoxin Concentrations in the Complete Feed for Farm Animals During 2021–2024 Period

In 2021, the concentrations of total aflatoxins in the analyzed samples varied between 0.1 and 2.90 ppb, the highest value being found for pigs and sows. The highest values for FB and DON were also found in the feed of pigs and sows, 4.7 ppm FB, and 1.9 ppm DON, respectively, but as in the case of AFT, the average concentration values were similar for all three feed categories. The EC recommendation for DON in pigs feed is 0.9 ppm, and the results showed that in 9.75% of complete feed samples for pigs and sows and 6.25% of feed samples for piglets and gilts, this recommendation has been exceeded. Moreover, analyzing the range values, the highest concentration distribution was observed in ZEA (1.9–62.80 ppb), T2/HT2 (2.8–32.1 ppb), and OTA (0.7–19.7 ppb), but without exceeding EC 1319/2016 recommendations.

In 2022 the distribution of concentration values was uniform for AFT, FB, and DON, the largest variations in the ranges being recorded for ZEA (0.7–32.1 ppb), T2/HT2 (0.1–11.9 ppb) and OTA (0.2–8.5 ppb). As in 2021, there were a few samples that exceeded EC recommendation in the case of DON (0.9 ppm) where the maximum concentration was 1.3 ppm, but only 2% of complete feed for pigs and sows, respectively, 5.25% for piglets and gilts exceeded the recommendation.

The range of concentration values recorded in 2023 for AFT (0.3–2.4 ppb), FB (0.7–4.7 ppm) and DON (0.2–1.9 ppm) were similar to those presented in previous years, the highest being again registered for ZEA (0.9–32.1 ppb), T2/HT2 (0.1–23.6), and OTA (0.2–19.7). DON was the only mycotoxin that exceeded EC recommendations in pigs’ and sows’ complete feed in 5.35% and 7.14% in piglets’ and gilts’ complete feed.

As in previous years, in 2024 the only mycotoxin exceeding the EC recommended levels was DON. In complete feed for pigs and sows, 1.6% of samples exceeded the 0.9 ppb level, while in piglets and gilts complete feed, the percentage is higher, with 6.7% of samples exceeding the recommendation.

Analyzing the annual evolution of the contamination level with each mycotoxin, we noticed that for pigs and sows complete feed, the average concentration of AFT was constant over the four years, but for FB, DON, ZEA, T2/HT2 the highest concentrations were recorded in 2021, while for OTA the highest concentration was in 2024. FB (*p* < 0.0001), DON (*p* = 0.0017, *p* = 0.0318), ZEA (*p* = 0.0004, *p* = 0.0245), T2/HT2 (*p* < 0.0001), and OTA (*p* = 0.0002, *p* = 0.0316), decreased significantly in 2022 and 2023 compared to 2021, T2/HT2 (*p* = 0.0032) and OTA (*p* < 0.0001) levels significantly increased again in 2024 (Figure 1).

In complete feed for piglets, the concentration levels of AFT and DON do not suffer statistically significant changes during the four-year studied period, however, the highest concentrations of FB, ZEA, and T2/HT2 were recorded in 2021, while in 2022, 2023, and 2024 the concentrations dropped significantly. For OTA the highest levels were observed in 2024, where the concentrations were significantly increased in comparison with all previous years (Figure 2).

As in the case of the complete feed for pigs, piglets, and poultry, the concentrations of AFT remain constant during the four-year period, the highest values of all the other mycotoxins (FB, DON, ZEA, T2/HT2, and OTA) being recorded in 2021, followed by a significant decrease in 2022, where the lowest values were observed (Figure 3). During 2023 and 2024 increased values were observed once again, but the maximum recorded values remain in 2021.

### 2.3. Mycotoxins Correlations in the Complete Feed for Farm Animals’ Samples During 2021–2024 Period

Analyzing the correlations between all six mycotoxins over the four-year period (Figure 4), we observed that for complete feed for pigs and sows the strongest and constant correlation is that between DON and ZEA, followed by AFT-DON and FB-OTA. In 2021, positive and significant correlations were observed for AFT-DON (r = 0.46) and AFT-OTA (r = 0.25), while in 2022 positive correlations were between DON and ZEA (r = 0.46), DON-T2/HT2 (r = 0.38) and ZEA-T2/HT2 (r = 0.63). In 2023, strong correlations were observed between ZEA and T2/HT2 (r = 0.65), AFT and DON (r = 0.52), DON and ZEA (r = 0.49), and DON and T2/HT2 (r = 0.40). As in previous years, in 2024 positive correlations existed for AFT-DON (r = 0.32), DON-ZEA (r = 0.42), DON-T2/HT2 (r = 0.43), and ZEA-T2/HT2 (r = 0.63).

Unlike complete feed for pigs and sows, where the strongest correlation over the four-year period was between DON and ZEA, in complete feed for piglets and gilts (Figure 5) the strongest correlation was between ZEA and T2/HT2, followed by AFT-FB and FB-OTA. In 2021, correlations between FB-T2/HT2 (r = 0.57) and AFT-FB (r = 0.52) were the strongest. A positive correlation between ZEA and T2/HT2 (r = 0.17) was also observed. Similarly, in 2022, correlations between AFT and FB (r = 0.37) and ZEA and T2/HT2 (r = 0.76), but a strong correlation between FB-OTA (r = 0.57) was noticed. The same AFT-FB (r = 0.21), ZEA-T2/HT2 (r = 0.23) and FB-OTA (r = 0.55) positive correlations were present in 2023, while in 2024, in addition to ZEA-T2/HT2 (r = 0.79) and FB-OTA (r = 0.36) correlations, AFT-ZEA (r = 0.7), AFT-T2/HT2 (r = 0.37) and DON-ZEA (r = 0.35) were also correlated.

During the 2021–2024 period, many positive correlations concerning the occurrence of mycotoxins in the complete feed of poultry were observed (Figure 6). Similarly with the feed for pigs and sows, but also for piglets and gilts, a positive DON-ZEA, respectively, ZEA-T2/HT2 correlation was found also in feed for poultry. Moreover, DON-T2/HT2, FB-T2/HT2, and FB-OTA were also positively correlated. In 2021, positive correlations were noticed for DON-ZEA (r = 0.12), DON-T2/HT2 (r = 0.14), ZEA-T2/HT2 (r = 0.24), and FB-T2/HT2 (r = 0.36). In the same line, in 2022, there were positive correlations between DON and ZEA (r = 0.3), DON-T2/HT2 (r = 0.43), ZEA-T2/HT2 (r = 0.69), but also FB-OTA (r = 0.22). In addition to the DON-ZEA (r = 0.29), DON-T2/HT2 (r = 0.46), ZEA-T2/HT (r = 0.59), and FB-OTA (r = 0.66) correlations, in 2023 there were also positive FB-T2/HT2 correlations (r = 0.65), with all these positive correlations being present also in 2024.

## 3. Discussion

The mycotoxins occurrence is a worldwide issue throughout the feed and food chain. Livestock farming is a particularly important economic sector that can be affected by the presence of these fungal secondary metabolites in cereal crops [30].Our work focuses on analyzing the concentrations of these contaminants and their occurrence in complete feed for farm animals produced in a local factory (pilot station of the National Institute of Biology and Animal Nutrition, Balotesti) from the southern region of Romania, including raw materials (corn, soybeans, wheat, barley, etc.) purchased, from the same region during the period 2021–2024. Constant monitoring of complete feeds is particularly important because they are particularly vulnerable to multiple contaminations, as they contain a mixture of several raw materials [5].

Our results showed that although in low concentrations, a minimum of four mycotoxins were detected simultaneously in complete feed for swine and poultry irrespective of the year or type of complete feed studied. Moreover, in over 58% of the samples, all six groups of mycotoxins studied were detected.

AFT and ZEA were detected in all samples during the entire period between 2021 and 2024. However, analyzing their evolution every year, we have noticed changes regarding the co-occurrence of mycotoxins. In 2021, AFT, ZEA, and OTA were detected in all samples regardless of their type, while in 2022, AFT and ZEA were ubiquitous. In 2023, AFT, ZEA, and DON were present in all the samples, whereas in 2024 all the six studied mycotoxins, five tested positive in all the samples, some of them being negative for T2/HT2 toxins.

Also, corroborating the analysis of the concentrations with EC guidance values, even in low percentages (<10%), we observed an exceedance of the recommended values of DON for some pigs’ complete feed samples.

A constant DON-ZEA association was observed in complete feed for pigs and sows throughout the 2021–2024 period, and ZEA-T2/HT2 for complete feed for piglets and gilts. Over the course of four years, three positive correlations were found regarding the complete feed for poultry: DON-ZEA, DON-T2/HT2, and ZEA-T2/HT2.

The highest concentrations for all mycotoxins studied were observed in 2021, but although these concentrations decreased in the following years, their presence in complete feed over time could imply serious health problems since the effects of mycotoxin co-occurrence are not yet known.

In Romania, there are not many statistical data on the occurrence of mycotoxin contamination in the feed of farm animals. Among the few existing studies from the southeast region of Romania, priority was given to the contamination of raw materials and less to finished products. A survey realized by Tabuc et al. reported that during 2008–2010, the most widespread mycotoxin in cereals from Southern Romania was DON (>70), while AFB1, ZEA, OTA, and FB were present in small percentages, below 40% [31]. By contrast, another study by Gagiu et al. (2018) focusing on post-harvest mycotoxin contamination of crops from the same area of (Southern) Romania from 2011 to 2015 observed a contamination with AFT and OTA [32]. Seeing the differences between the two studies over time, we can state again that permanent monitoring of mycotoxin concentrations and their reporting is absolutely necessary. Compared to other EU countries, it was reported that in Bulgaria, a country neighboring Romania, in 2017 [33] and 2021 [34] barley and wheat crops were mostly contaminated with OTA and ZEA, other mycotoxins occurrences like DON, FB, and AFT being low. In Hungary in 2016, a high detection rate of DON and ZEA, but also T-2 in swine feed was reported [35]. Their presence can be correlated with the data of another study showing that in 2015 in Hungary, corn crops, the main ingredient in pig feed, were contaminated in 86% of cases with DON, respectively, 41% ZEA and 52% T2 [36]. Similarly, in Poland, in the period 2015–2020, the presence of DON and ZEA was reported in over 99% of over 2000 analyzed feeds, in some cases the exceedance of the recommendations of the European Commission was observed [37]. The contamination of pig feed with DON and ZEA in higher percentages was also reported in Croatia, the presence of AFB1, OTA, and FB was also reported, but in lower ratios [38]. Corroborating this information, in the areas neighboring Romania it seems that the common mycotoxin is ZEA, the occurrence of the other mycotoxins detected varies probably due to geographical, climatic, or soil conditions. However, there are also countries such as Albania, which in 2022 reported a significant occurrence of AFB1, this being present in over 88% of corn crops [39].

If in countries geographically close to Romania, a high incidence of DON, OTA, and ZEA was observed, in extreme European countries such as Portugal, the highest detection rate reported was for OTA and FB [40]. For ZEA the detection rate was 25%, by comparison with our results that show the presence of ZEA in 100% of complete feed samples from Southeastern Romania. In Spanish barley, the simultaneous presence of at least two mycotoxins was reported, the majority being combinations of three or four mycotoxins (AFB1, DON, OTA, and AFB1, DON, OTA, ZEA), but regarding the complete feed, there are not much data available [41]. The presence in this study of at least four mycotoxins in the complete feed of farm animals represents a major concern, although their concentrations are lower than the EC recommendations.

It is known that the climate is an important factor in the development of fungi, and changes in temperature and humidity, along with processing and storage conditions, play an important role in the production of mycotoxins. This is probably one of the reasons why the presence and level of contamination varies from one country to another. Long-term studies are needed to monitor the mycotoxin occurrence in farm animal feed. This kind of study could provide predictability, necessary to combat or mitigate the problem of contamination with mycotoxins, which could be potentiated by many environmental factors, including climate changes [41].

## 4. Conclusions

The present study provides data on mycotoxin contamination of complete feeds for swine and poultry available in the local market of the southern area of Romania. Our results show that the most frequent mycotoxins detected were AFT and ZEA, but also DON, OTA, T-2/HT2, and FB were detected in more than 85% of samples.

During 2021–2024, a constant and statistically significant DON-ZEA correlation was observed in complete feed for pigs and sows as well as for ZEA-T2/HT2 in complete feed for piglets and gilts, evidence of co-occurrence of these mycotoxins. Regarding complete feed for poultry three positive correlations were observed over the four-year period DON-ZEA, DON-T2/HT2, respectively, ZEA-T2/HT2.

Our goal is to continue monitoring the mycotoxin presence in the complete feed of farm animals from southern Romania and to provide a starting point for future studies that could offer solutions to counteract their toxic effects, which can lead to an improvement in the life quality of animals.

## 5. Materials and Methods

In the present study, several complete feeds produced in a local factory from Romanian Wallachian Plain during a four-year period (2021–2024) and available in local market of south region of Romania were analyzed by ELISA (enzyme-linked immunosorbent assay), used as commonly method for mycotoxins determination in cereals. Veratox (Neogen, Lansing, MI, USA) kits which provide a competitive direct ELISA to obtain exact concentration of mycotoxins were used.

### 5.1. Sample Collection

A total of 596 analyzed samples, of which 206 pigs’ and sows’ complete feed, 79 piglets’ and gilts’ complete feed, and 311 poultry complete feed were analyzed (Table 2). Every sample was collected in accordance with Directive (EU) 2023/2782 of the European Commission, which regulates the number of subplots and increments depending on the size of the batch produced, with the aggregate sample weight varying between 1 and 10 kg.

### 5.2. Sample Preparation

For the determination of AFT, FB, ZEA, T2/HT2, methanolic extracts were prepared. For each sample, 5 gr of ground feed was weighed and mixed, according to manufacturer instructions in 1:5 ratio, with 25 mL of 75% methanol (MetOH) solution. The obtained mixture was stirred for 15 min and then passed through Whatman Grade 1 cellulose filters. Similarly, the extracts for the determination of DON and OTA were obtained, but the solvents used for their extractions were water, respectively, 50% MetOH solution.

### 5.3. Total Aflatoxins Determination

For all the samples, the concentration of total aflatoxins (B1, B2, G1, G2) was determined using the Veratox (Neogen, Lansing, MI, USA) high-sensitivity kit for quantitative aflatoxins determination, according to manufacturer instructions. Briefly, 100 µL of each sample was mixed with 100 µL of AFB-HRP conjugate solution provided by the kit, placed in the antibody-coated microwells, and incubated at room temperature for 2 min. Afterward, the plate was washed 5 times with deionized water, and 100 µL of K-Blue Substrate solution was added and incubated for 3 min. The reaction is stopped with 100 µL of Red Stop Solution provided by the kit, and then the absorbance is read at a wavelength of 650 nm. Concentrations were calculated using a standard curve obtained by several dilutions of aflatoxin standards and Neogen Veratox software version 3.0. All the samples were analyzed in triplicate, the results expressed being a media between those individual determinations. The limit of detection e of AFT concentration by this immunoenzymatic method is 0.4 ppb.

### 5.4. Fumonisins Determination

Fumonisins (B1, B2, B3) were also determined by the ELISA method, using the Veratox kit (Neogen, Lansing, MI, USA) for fumonisin quantitative determination, according to manufacturer instructions. Similar to AFB determination, 100 µL of each sample was mixed with 100 µL of FB-HRP conjugate solution, placed in antibody-coated microwells, and incubated for 10 min at room temperature. After 5 washes with deionized water, 100 µL of K-Blue Substrate solution was added, following a 5 min incubation. Stopping the reaction with 100 ul Red Stop Solution was followed by reading the absorbance of the samples at 650 nm. A standard curve derived from multiple dilutions of fumonisins standard and Neogen Veratox software were used to calculate concentrations. All the samples were analyzed in triplicate, the results expressed being a media between those individual determinations. The specific limit of detection for FB concentration is 0.2 ppm.

### 5.5. Deoxynivalenol Determination

The concentration of DON for each sample was determined with the Veratox kit (Neogen, Lansing, MI, USA) for DON 2/3 quantitative determination. A 100 µL amount of each sample and 100 µL of DON-HRP conjugate solution were mixed and placed in antibody-coated wells and incubated for 2 min at room temperature. At the end of the incubation, the plates were washed 5 times with deionized water, and 100 µL of K-Blue^®^ Substrate solution was added and incubated for 3 min. After stopping the reaction, the absorbance of the samples was read at 650 nm. Concentrations were calculated using a standard curve obtained by several dilutions of deoxynivalenol standards and Neogen Veratox software. All the samples were analyzed in triplicate, the results expressed being a media between those individual determinations. The detection limit of DON concentration by this immunoenzymatic method is 0.1 ppm.

### 5.6. Zearalenone Determination

The concentration of ZEA in each sample was determined using the Veratox kit (Neogen, Lansing, MI, USA) for zearalenone quantitative determination. After a 5 min incubation of 100 µL of the sample and 100 µL of ZEA-HRP conjugate solution, followed by 5 washes with deionized water, the addition of 100 µL of K-Blue^®^ Substrate solution and a 5 min incubation, the reaction was stopped with 100 µL of Red Stop Solution, and the absorbance of the samples was read at 650 nm. A standard curve derived from multiple dilutions of zearalenone standards was used to calculate the concentrations. All the samples were analyzed in triplicate, the results expressed being a media between those individual determinations. The detection limit of this kit for ZEA concentration is 5 ppb.

### 5.7. T2/HT2 Determination

The concentration of T2 and HT2 toxins in each sample was determined using the Veratox kit (Neogen, Lansing, MI, USA) for quantitative T-2/HT-2 determination. According to the manufacturer’s instructions, the protocol steps for their determination are similar to those for determining the ZEA concentration, but the limit of detection is different: for T2 and HT2 the threshold starts from 7.5 ppb. Concentrations were calculated using a standard curve obtained by several dilutions of T2/HT2 control and Neogen Veratox software. All the samples were analyzed in triplicate, the results expressed being a media between those individual determinations.

### 5.8. Ochratoxins Determination

Considering its widespread cereals, the concentration of ochratoxins (A, B) in feed was also determined, using the Veratox kit (Neogen, Lansing, MI, USA) for quantitative ochratoxin kit. Briefly, 100 µL of each sample was mixed with 100 µL of ochratoxin HRP conjugate solution and incubated for 10 min. After 5 washes, 100 µL of K-Blue Substrate was added in each well, followed by a 10 min incubation. Once the reaction was stopped with Red Stop Solution, as in the case of the other studied mycotoxins, the absorbance was read at 650 nm. A standard curve that was created by dilutions of ochratoxins standards was used to calculate concentrations. All the samples were analyzed in triplicate, the results expressed being a media between those individual determinations. The specific detection limit of Veratox for ochratoxin is 1 ppb.

### 5.9. Statistical Analysis

All results were graphically represented and statistically analyzed using Fischer’s exact test, two-way ANOVA, and Pair Panels function available on GraphPad Prism version 9.3.0 and R software version 4.3.2 (31 October 2023 UCRT). The statistical significance was marked graphically as follows: * significant (0.05 ≥ *p* > 0.01), ** distinct significant (0.01 ≥ *p* > 0.001), *** very significant (0.001 ≥ *p* > 0.0001) and **** extremely significant (*p* < 0.0001), ns—not significant (*p* ≥ 0.05).

## Figures and Tables

**Figure 1 toxins-17-00201-f001:**
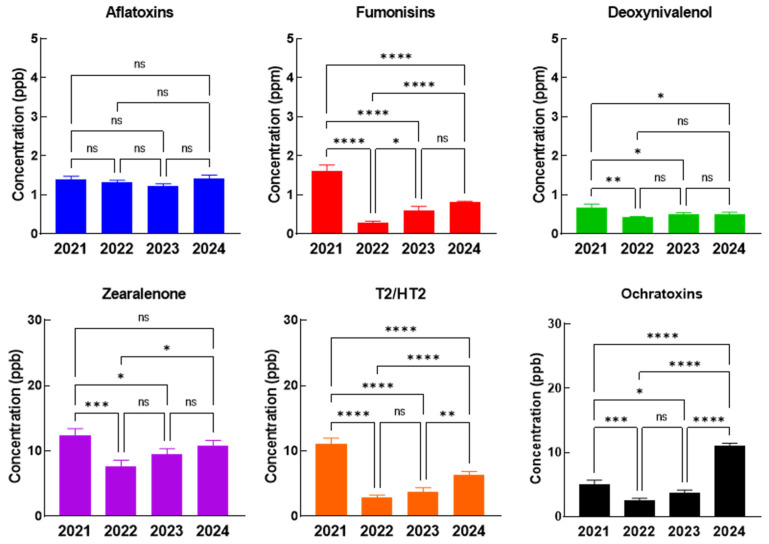
The evolution of mycotoxin concentrations in the complete feed of pigs and sows in 2021–2024. The statistical significance was marked graphically as follows: * significant (0.05 ≥ *p* > 0.01), ** distinct significant (0.01 ≥ *p* > 0.001), *** very significant (0.001 ≥ *p* > 0.0001) and **** extremely significant (*p* < 0.0001), ns—not significant (*p* ≥ 0.05).

**Figure 2 toxins-17-00201-f002:**
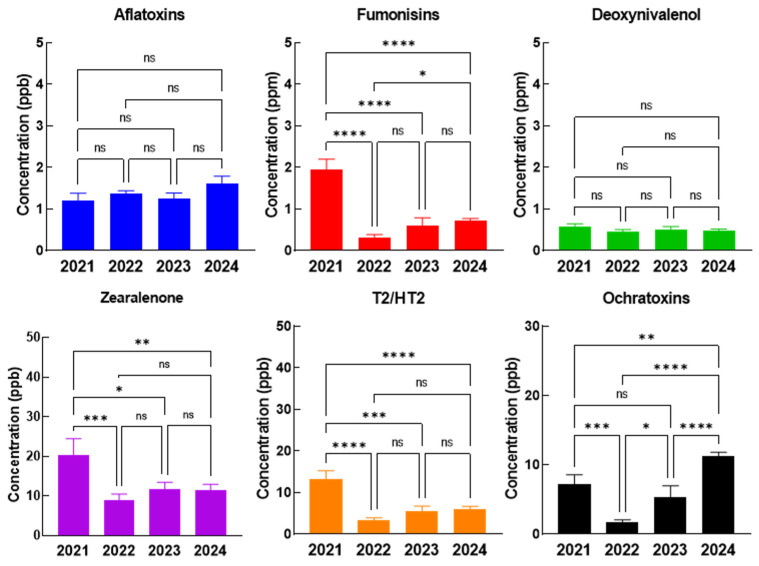
The evolution of mycotoxin concentrations in the complete feed of piglets and gilts in 2021–2024. The statistical significance was marked graphically as follows: * significant (0.05 ≥ *p* > 0.01), ** distinct significant (0.01 ≥ *p* > 0.001), *** very significant (0.001 ≥ *p* > 0.0001) and **** extremely significant (*p* < 0.0001), ns—not significant (*p* ≥ 0.05).

**Figure 3 toxins-17-00201-f003:**
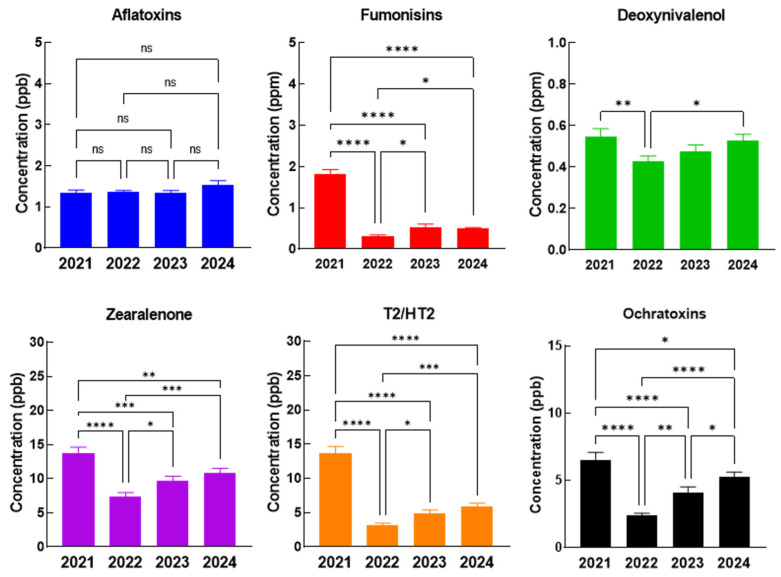
The evolution of mycotoxin concentrations in the complete feed of poultry in 2021–2024, The statistical significance was marked graphically as follows: * significant (0.05 ≥ *p* > 0.01), ** distinct significant (0.01 ≥ *p* > 0.001), *** very significant (0.001 ≥ *p* > 0.0001) and **** extremely significant (*p* < 0.0001), ns—not significant (*p* ≥ 0.05).

**Figure 4 toxins-17-00201-f004:**
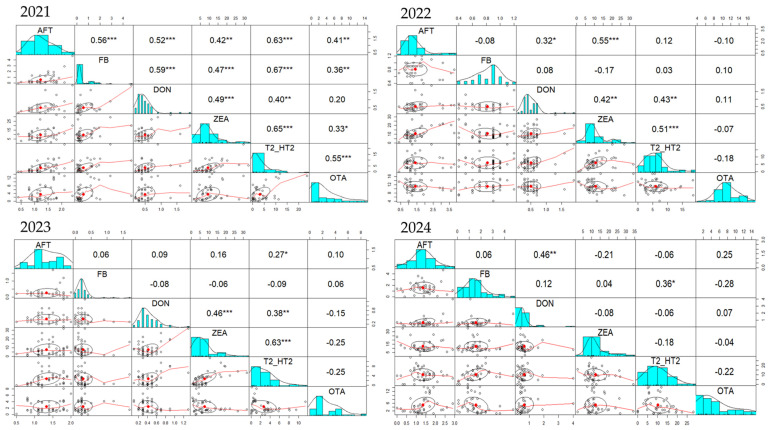
Pairs panel chart of Spearman correlation test regarding the occurrence of mycotoxins in complete feed of pigs and sows during 2021–2024 period. The statistical significance was marked graphically as follows: * significant (0.05 ≥ *p* > 0.01), ** distinct significant (0.01 ≥ *p* > 0.001) and *** very significant (0.001 ≥ *p* > 0.0001).

**Figure 5 toxins-17-00201-f005:**
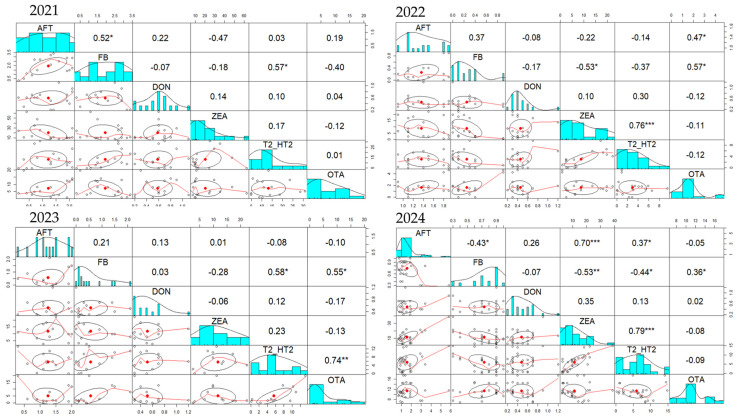
Pairs panel chart of Spearman correlation test regarding the occurrence of mycotoxins in complete feed of piglets and gilts during 2021–2024 period. The statistical significance was marked graphically as follows: * significant (0.05 ≥ *p* > 0.01), ** distinct significant (0.01 ≥ *p* > 0.001) and *** very significant (0.001 ≥ *p* > 0.0001).

**Figure 6 toxins-17-00201-f006:**
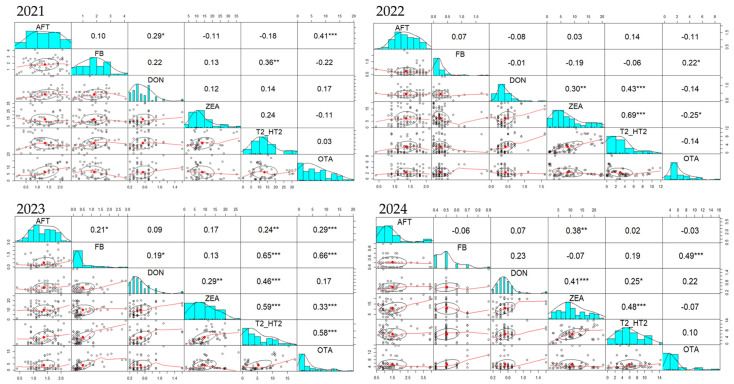
Pairs panel chart of Spearman correlation test regarding the occurrence of mycotoxins in complete feed of poultry during 2021–2024 period. The statistical significance was marked graphically as follows: * significant (0.05 ≥ *p* > 0.01), ** distinct significant (0.01 ≥ *p* > 0.001) and *** very significant (0.001 ≥ *p* > 0.0001).

**Table 1 toxins-17-00201-t001:** Sample contamination with mycotoxins in complete feed for farm animals’ samples during 2021–2024 period.

			AFT	FB	DON	ZEA	T2/HT2	OTA
Pigs and sows complete feed	2021	Samples	41/42	41/42	41/42	42/42	39/42	42/42
		%	97.62	97.62	97.62	100	92.86	100
	2022	Samples	55/55	49/55	55/55	55/55	52/55	52/55
		%	100	89.09	100	100	94.55	94.55
	2023	Samples	56/56	54/56	56/56	56/56	48/56	54/56
		%	100	98.21	100	100	85.71	96.43
	2024	Samples	53/53	53/53	53/53	53/53	51/53	53/53
		%	100	100	100	100	96.22	100
Piglets and gilts complete feed	2021	Samples	16/16	16/16	16/16	16/16	16/16	16/16
		%	100	100	100	100	100	100
	2022	Samples	19/19	15/19	19/19	19/19	17/19	17/19
		%	100	78.95	100	100	89.47	89.47
	2023	Samples	14/14	13/14	14/14	14/14	14/14	13/14
		%	100	92.86	100	100	100.00	92.86
	2024	Samples	30/30	30/30	30/30	30/30	30/30	30/30
		%	100	100	100	100	100	100
Poultry complete feed	2021	Samples	63/63	63/63	63/63	63/63	61/63	63/63
		%	100	100	100	100	96.83	100
	2022	Samples	108/108	89/108	106/108	108/108	102/108	102/108
		%	100	83.33	98.15	100	94.44	94.44
	2023	Samples	75/75	68/75	75/75	75/75	73/75	73/75
		%	100	90.67	100	100	97.33	97.33
	2024	Samples	65/65	65/65	65/65	65/65	64/65	65/65
		%	100	100	100	100	98.46	100

**Table 2 toxins-17-00201-t002:** The distribution by year and type of the studied complete feed samples.

Year	Complete Feed for Pigs and Sows (No. Samples)	Complete Feed for Piglets and Gilts (No. Samples)	Complete Feed for Poultry (No. Samples)
2021	42	16	63
2022	55	19	108
2023	56	14	75
2024	53	30	65
Total	206	79	311
	596 samples		

## Data Availability

The original contributions presented in this study are included in the article. Further inquiries can be directed to the corresponding author.

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
