# Peer review of "Trends in Mycotoxins Co-Occurrence in the Complete Feed for Farm Animals in Southern Romania During 2021–2024 Period"

_toxins, 2025, doi:10.3390/toxins17040201_

Round 1

Reviewer 1 Report

Comments and Suggestions for Authors

Please refer to attached.

Author Response

Thank you for the author’s hard work, and I think your paper was implemented to fit your research purpose. My review is based on the author’s manuscript. My comments are below.

  1. Depending on the growth and reproduction classification, such as pigs and sow or pigs and gilts, there is a possibility that the degree and characteristics of contamination with mycotoxins may vary depending on the composition and ratio of raw materials that become pollutants of mycotoxins in the complete feed.

We totally agree with your comment. Indeed, for each category considered in our study (pigs, sows, piglets and gilts) there are different nutritional requirements according to the nutritional value tables (eg. NRC) for different categories (eg. age, sex etc). However, regardless of the nutritional formula, corn/wheat and soybeans represent the largest percentage in the diets (>85%), and the contamination of these raw materials with mycotoxins subsequently causes contamination of the complete feed.

  1. You described in line 25-26 further nutritional studies are need to establish limit of the contamination level for mycotoxins for which there are no regulations, but in order to establish maximum residue limit(MRL) for mycotoxins, risk assessment through monitoring rather than nutritional research must be conducted.

Thank you for your valuable comment. Our study was related to the assessment of the occurrence of five most common mycotoxins (AFT, FB, DON, ZEA, T2/HT) in different feed samples (swine and poultry samples). The interpretation of our results was done according to the EC in force legislation that have established (for AF) or recommend (for the other mycotoxins) guidance values for feed as presented in Table 3.  The present study doesn’t contain any nutritional study. As suggested, we have replaced the line 25-26 with the following phrase in the Introduction section:

“Our survey study can provide important data showing the degree of contamination of pig and poultry feed with mycotoxins including the simultaneous presence of different mycotoxins in these complete feed”

  1. In order to understand the contents in Figure 1, it seems necessary to explain ns, *~****.

Thank you for your suggestion. We have changed the Legend for the figures 1, 2 and 3 in the new version of the manuscript by adding the following explanation:

 “The statistical significance was marked graphically as follows: *significant (0.05 ≥ p > 0.01) , **very significant (0.01  ≥  p > 0.001) and ***extremely significant (0.001  ≥  p > 0.0001)”.

  1. In accordance with the contents of lines 226 to 228, the year in the title of Figure 7 should be revised from 2024 to 2023, and the year in the title of Figure 8 should also be revised from 2022 to 2024.

Thank you for your remarks, we have made the suggested correction.

  1. Including the Table 2, 4, 6, and 8, the ppb unit and result concentration of T2/HT2 in this manuscript is confused whether it is right or not. The reason is that you described in line 442 to 443 that for T2 and HT2 the threshold starts from 10 ppm. And if dilution factor in sample preparation is considered, the detection limit of T2/HT2 starts from 2 ppm.

Thank you for your remarks, the unit is ppb, we made the necessary changes in line 442 and 443. The results obtained took into account the dilution factor, and all the specifications mentioned in this manuscript for each kit are provided by the manufacturer.

  1. In line 400, why did you use 5g of the sample for determination of mycotoxins, and how many repetitions do you have?

All samples were processed following the protocol provided by the manufacturer where the recommended dilution is 1:5 in MetOH solution. We added this information in the manuscript. All the samples are analyzed in triplicate, the results expressed being a media between those individual determinations.

Reviewer 2 Report

Comments and Suggestions for Authors

Introduction

You should briefly introduce animal breeding of seed consumption in your country, or this research is of little significance. Every paragraph should have its main point you want to state, combine small paragraphs.

Materials and Methods

The weight of every sample. The distribution of market. The number of sample in each location.

Results

You should rewrite this part. This section is too long.

Detailed Introduction of every toxin in every year is unnecessary and tedious. Just mention the main result, or it is uncomfortable for the readers to get the useful information. The high positive rate is similar in four years. The values in table 1 and 2 can be listed in the text and combined in fig 1. One section for 2.1-2.7.

Fig 4, 9, 14 are unnecessary. List the important values in the text. Simplify the figures about pairs panel chart of Spearman correlation. The analysis in one year or about one kind of feed in a figure.

The occurrence of mycotoxins is related with the source, storage, sampling time, environment, and it is not easy to get a definitive conclusion between varied years. The analysis about different kinds of feed could be better.

Discussion

The first mycotoxin data about contamination in complete feed? Please check it.

In the front paragraphs, only the data in other reports were listed and no analysis about the difference. Nothing were discussed.

Geographic location of your country is background, so it should not be present in this part. No climate data in the result.  

Comments on the Quality of English Language

paragraph structure is terrible

Author Response

Introduction

  1. You should briefly introduce animal breeding of seed consumption in your country, or this research is of little significance. Every paragraph should have its main point you want to state, combine small paragraphs.

Thank you for your kind remark we added the lines:

” The feed categories were selected due to their economic importance, according to Eurostat. Accordingly, Romania has a pig population of 3.27 million, 637,000 piglets and approximately 7.5 million chicks of mixed meat laying. The main cereals (purchased from local feed producers) used as the raw materials of all complete feed formulations were corn (>65%) and soybean meal (>19%), the degree of variation in the contamination level of the analyzed feeds depending on the percentage of cereals in the feed formulation calculated to ensure the nutrient requirements for the species and age category of the farm animals.

We also combined some of the smaller paragraphs and modified them through the entire introduction part.

Materials and Methods

  1. The weight of every sample. The distribution of market. The number of samples in each location

Thank you for your kind remark. The number of batches and the sample weights varied depending on the lot weight produced in the factory. In order to offer more details about the samples collection procedure we added in the materials and method the phrase: “Every sample was collected in accordance with Directive (EU) 2023/2782 of the European Commission, which regulates the number of sublots and increments depending on the size of the batch produced, the aggregate sample weight varying between 1-10 kg”.

The complete feed subjected to the mycotoxin analyze were produced in a local factory (pilot station of the National Institute of Biology and Animal Nutrition) from the southern region of Romania from raw materials (corn, soybeans, wheat, barley, etc) purchased, from the same region. The distribution on the market is related to the southern region of Romania, more precisely counties like ARGES, DAMBOVITA, PRAHOVA, TELEORMAN, GIURGIU, CALARASI, IALOMITA.

We also added more details about the production process in the lines 94-103. “Therefore, the aim of our research was to provide data on occurrence and concentration levels of six major mycotoxins (AFT, FB, DON, ZEA, T2/HT and OTA)  that are most found in poultry, piglets, and pigs complete feed produced in a local factory (pilot station of the National Institute of Biology and Animal Nutrition) from the southern region of Romania and their co-occurrence over a four-year period (2021–2024). The feed categories were selected due to their economic importance. The feed analyzed was selected based on the demand on the feed market and the economic importance of the farm animal species for which they were manufactured. According to Eurostat (European Statistical System), Romania has a pig population of 3.27 million, of which 637,000 are piglets [27] and approximately 7.5 million chicks of mixed meat laying breeds [28]. The main cereals (purchased from local feed producers) used as the raw materials of all complete feed formulations were corn (>65%) and soybean meal (>19%), the degree of variation in the contamination level of the analyzed feeds depending on the percentage of cereals in the feed formulation calculated to ensure the nutrient requirements for the species and age category of the farm animals.”

Results

  1. You should rewrite this part. This section is too long.

Detailed Introduction of every toxin in every year is unnecessary and tedious. Just mention the main result, or it is uncomfortable for the readers to get the useful information. The high positive rate is similar in four years. The values in table 1 and 2 can be listed in the text and combined in fig 1. One section for 2.1-2.7.

Thank you for your kind remarks, as recommended we rearranged and redistributed all the results.

  1. Fig 4, 9, 14 are unnecessary. List the important values in the text. Simplify the figures about pairs panel chart of Spearman correlation. The analysis in one year or about one kind of feed in a figure.

Thank you for your valuable contribution, we deleted fig 4,9,14, and we also merged with the Spearman correlation figures as you suggested.

  1. The occurrence of mycotoxins is related with the source, storage, sampling time, environment, and it is not easy to get a definitive conclusion between varied years. The analysis about different kinds of feed could be better.

The analysis of mycotoxin contamination of different types of raw materials is very important and many studies have addressed this analysis. Raw materials from different source, storage, sampling time, environment, enter in different percentages in complete feed formulations and for this reason we consider that to analyze the entire complete feeds is just as important because ultimately the animals receive the complete feed for eating.

Discussion

  1. The first mycotoxin data about contamination in complete feed? Please check it.

Thank you for your remark, we changed the phrase that contained this information into ” The present study provides data on mycotoxin contamination of complete feeds for swine and poultry available on local market of southern area of Romania”

  1. In the front paragraphs, only the data in other reports were listed and no analysis about the difference. Nothing were discussed.

Thank you for your remark, we added more lines in which we discussed our result ( see lines 285-307)

  1. Geographic location of your country is background, so it should not be present in this part. No climate data in the result. 

Thank you for your valuable observation, we deleted those paragraphs

Reviewer 3 Report

Comments and Suggestions for Authors

General comments
This paper addresses an interesting topic concerning mycotoxin loads in feeds collected in Romania. Co-occurrence already includes occurrence. The authors should not used simultaneaously these two words. 

New title for this manuscript is recommended.... "Trends in mycotoxns co-occurrence in....".could be one alternative title.

The study has been performed in a routinely analyses basis over four years without any quality assessment. No use of reference materials or quality control for targeted mycotoxins in cereal or feed. No validation data were given to prove the quality of the analytical method used. ELISA method required a confirmation method.
Figures 5, 6, 7, and 8 should be changed. They are not appropriates, as presented.

Figures 10, 11, 12, 13, 15, 16, 17, and 18 should be changed. They are not appropriates, as presented.  

Specific comments
Line 37 and line 93-94: We understood that "food" means humans' food
and feed means animal feed.  so short this sentence and throughout the manuscript.
Live 40: EU regulations should be mentionned in this paragrapgh and throughout the manuscript. The reference section  

Throughout all the manuscript, please carefully check all figures and percentages and round them with one decimal place “See table 1: 97.6 % instead of 97.62 %....;" (see Table 2, 3 and 4 also).
Line 79, space between coccisidios and reference 22
Line 98: state T2/HT2 instead of t2/HT.
Sample collection did mentionne the amounts of collected batches 

Line 406: Total Aflatoxins....remove capitalized letters a

For the methods and materials, remove also the capitalized letters

With regards to the international requirements for analytical validation, this manuscript did not present the main characteristics of the used analytical methods. So that I can not recommend its publication.

Comments on the Quality of English Language

Not checked (in details)

Author Response

General comments
1. This paper addresses an interesting topic concerning mycotoxin loads in feeds collected in Romania. Co-occurrence already includes occurrence. The authors should not used simultaneaously these two words. 

Thank you very much for your kind remarks, we revised the manuscript and we eliminated the simultaneous use of co-occurrence and occurrence.

  1. New title for this manuscript is recommended.... "Trends in mycotoxns co-occurrence in....".could be one alternative title.

Thank you for your kind suggestion, as you recommended, we changed the title in “Trends in mycotoxins co-occurrence in the complete feed for farm animals in southern Romania during 2021-2024 period”

  1. The study has been performed in a routinely analyses basis over four years without any quality assessment. No use of reference materials or quality control for targeted mycotoxins in cereal or feed. No validation data were given to prove the quality of the analytical method used. ELISA method required a confirmation method.

Thank you for your valuable remark. We used ELISA method as a commonly method for the control of the levels of mycotoxins in cereals. For the purpose of mycotoxin screening, a certainty of 95 % is considered fit-for-purpose (Commission Regulation (EU) No 519/2014 of 16 May 2014). The Veratox ELISA kits that we used for this study fulfill this criterion containing pure toxin standards, the mycotoxin concentration being calculated by reference to these standards. We also periodically perform random confirmation tests by HPLC. We found ELISA method used in many articles with mycotoxins survey

  • Microbiological and Mycotoxicological Quality of Common Wheat in Romania in the Extremely Dry 2023–2024 Agricultural Year
  • DEOXYNIVALENOL-AND-HEAVY-METALS-CONTAMINATION-IN-COMMON-WHEAT-IN-ROMANIA-IN-THE-EXTREMELY-DRY-YEAR-2015.pdf
  • TOTAL-AFLATOXIN-CONTAMINATION-IN-COMMON-WHEAT-IN-ROMANIA-IN-THE-YEARS-2015-AND-2016-WITH-EXTREME-WEATHER-EVENTS.pdf
  • Detection of mycotoxins (aflatoxins) in primary and processed foods for humans and farm animals, in Riobamba-Ecuador. org/doi/full/10.5555/20230506246
  • Aflatoxin levels in fish feeds in Abeokuta, Ogun State Description: Aflatoxin levels in fish feeds in Abeokuta, Ogun State
  • Aflatoxin Sampling and Testing Proficiency in the Texas Grain Industry View of Aflatoxin Sampling and Testing Proficiency in the Texas Grain Industry
  • (PDF) Ochratoxin A and Aflatoxins: Fine-Tuning to the ELISA Test on Table Olives
  • Quantification of Ochratoxin A in 90 spice and herb samples using the ELISA method - PMC
  • Mycotoxins in red wine: Occurrence and risk assessment - ScienceDirect
  • Validation Study of a Rapid ELISA for Detection of Deoxynivalenol in Wheat, Barley, Malted Barley, Corn, Oats, and Rice | Journal of AOAC INTERNATIONAL | Oxford Academic

  1. Figures 5, 6, 7, and 8 should be changed. They are not appropriates, as presented.

Figures 10, 11, 12, 13, 15, 16, 17, and 18 should be changed. They are not appropriates, as presented.  

Thank you for your suggestion, we compressed the figures, and we also tried to simplify all the data.

Specific comments
5. Line 37 and line 93-94: We understood that "food" means humans' food
and feed means animal feed.  so short this sentence and throughout the manuscript.

Thank you for your remark, we changed it in “in food and feed”

  1. Live 40: EU regulations should be mentioned in this paragraph and throughout the manuscript. The reference section  

Thank you for your kind suggestion, we added the regulations/recommendations in the new version of the manuscript: “with the exception of aflatoxins (EC 574/2011), for the other mycotoxins there are only recommendations (EC 1319/2016) regarding the maximum levels of mycotoxins allowed in feed for farm animals.”

  1. Throughout all the manuscript, please carefully check all figures and percentages and round them with one decimal place “See table 1: 97.6 % instead of 97.62 %....;" (see Table 2, 3 and 4 also).

Thank you, we revised this aspect, and in order to see the differences between the complete feed type and also through the years we merged and rearrange Tables 1-8.

  1. Line 79, space between coccisidios and reference 22

Thank you, we added the spaces, and we also checked once again this aspect all over the manuscript.

  1. Line 98: state T2/HT2 instead of t2/HT.
    Sample collection did mentioned the amounts of collected batches 

Thank you for your kind remark. The number of batches and sample weights varied depending on the lot weight produced in the factory. As recommended more details about the samples collection procedure was added in the materials and method of the revised version:

“Every sample was collected in accordance with Directive (CE) 2023/2782 of the European Commission, which regulates the number of sublots and increments depending on the size of the batch produced, the aggregate sample weight varying between 1-10 kg.”

  1. Line 406: Total Aflatoxins....remove capitalized letters a

For the methods and materials, remove also the capitalized letters

Thank you for your remark, we chanced A in a.
